# Nano-Liquid Chromatography with a New Monolithic Column for the Analysis of Coenzyme Q10 in Pistachio Samples

**DOI:** 10.3390/molecules28031423

**Published:** 2023-02-02

**Authors:** Cemil Aydoğan, Büşra Beltekin, Nurullah Demir, Bayram Yurt, Ziad El Rassi

**Affiliations:** 1Food Analysis and Research Laboratory, Bingöl University, Bingöl 12000, Türkiye; 2Department of Chemistry, Bingöl University, Bingöl 12000, Türkiye; 3Department of Food Engineering, Bingöl University, Bingöl 12000, Türkiye; 4Department of Chemistry, Oklahoma State University, Stillwater, OK 74078, USA

**Keywords:** coenzyme Q10, food analysis, miniaturization, monolith, nano-LC

## Abstract

Coenzyme Q10 (CoQ10) is a vital substance found throughout body. It helps convert food into energy and is eaten small amounts in foods. CoQ10 has gained great interest in recent years as a potential candidate for the treatment of various diseases. The content of CoQ10 in food samples is a crucial quality index for foods. Therefore, the development of sensitive separation and quantification method for determining the amount of CoQ10 in various samples, especially in foods, is an important issue, especially for food nutrition. In this study, a new, miniaturized monolithic column was developed and applied for the determination of CoQ10 in pistachio samples by nano-liquid chromatography (nano-LC). The monolithic column with a 50 µm i.d. was prepared by in situ polymerization using laurylmethacrylate (LMA) as the main monomer and ethylene dimethacrylate (EDMA) as the crosslinker. Methanol (MeOH) and polyethyleneglycol (PEG) were used as porogenic solvents. The final monolithic column was characterized by using scanning electron microscopy (SEM) and chromatographic analyses. The monolithic column with a 50 µm i.d. was applied to the analysis of CoQ10 in pistachio samples in nano-LC. This analytical method was validated by means of sensitivity, linearity, precision, recovery, and repeatability. The LOD and LOQ values were 0.05 and 0.48 µg/kg, respectively. The developed method using the monolithic column was optimized to achieve very sensitive analyses of CoQ10 content in the food samples. The applicability of the method was successfully demonstrated by the analysis of CoQ10 in pistachio samples.

## 1. Introduction

In the current global marketplace, several dietary supplements are easily available in food markets and on e-commerce platforms [1]. Among other dietary supplements, coenzyme Q10 (CoQ10), also known as 2,3-dimethoxy-5-methyl-6-proprenyl-1,4-benzoquinone or ubiquinone, is a endogenous, natural, lipophilic provitamin and redox active compound that has a very large and essential network in biological reactions. It is an effective natural antioxidant and has critical role in cellular bioenergetic cycles [2]. CoQ10 acts as the electron carrier for the respiratory chain in mitochondria and interacts with oxygen-derived radicals and singlet oxygen, preventing the initiation of lipid peroxidation and damage to biomolecules. In addition, it is known that CoQ10 has functions in ensuring membrane stability, gene expression, cell signaling, cell growth, and control of apoptosis. Meat (especially heart and liver), fish, nuts, peanuts, pistachios, and some oils are the main food sources that are richest in CoQ10 content, while much lower levels can be found in some dairy products, vegetables, fruits, and grains [3]. Scientific evidence regarding CoQ10 efficacy and safety is provided in the notable review article, [2]. There is also a large potential in this area of research, especially in metabolomics: in recent years, many developments have been observed in the analytical and spectroscopic methods of identification of the natural compounds (e.g., CoQ10) that can be present in fruits or vegetables. The rapid development in promising technologies, such as advanced chromatography and nuclear magnetic resonance (NMR), is reflected in new developments in instrumentation that improve resolution, sensitivity, and speed. These advances enable improved molecule detection and quantitation under in vivo as well as in vitro conditions [4,5,6,7]

The analysis of food products is of utmost importance, as foods include many ingredients consisting of naturally occurring compounds with nutritional value as well as contaminating substances [8]. Conventional LC is widely used for the analysis of CoQ10 in dietary supplements. For instance, Ercan and Nehir investigated the changes in the content of CoQ10 in various food samples using HPLC with a 4.6 mm i.d. reversed phase column [9]. Xue et al. developed an HPLC method for the analysis of CoQ10 in bee pollen samples [10]. However, the classical HPLC, with a 4.6 mm i.d. column, may be unsuitable for several reasons, such as a high chemical consumption, low sensitivity, and environmental contamination. Recent progress in LC has led to the use of miniaturized chromatographic separations [6,11]. This technology is now a reliable alternative to the classical HPLC in many areas of analysis. Among other miniaturized LC techniques, nano-LC is a promising tool for advanced food analysis [12,13]. The use of nano-LC in food analysis allows for several advantages, including great separation efficiency, a reduced consumption of chemicals, and green and advanced chromatography. This technique uses different type of columns, such as monolithic, particle-packed, open tubular, and pillar array, with internal diameters in the range of 5–100 μm [14]. Among these columns, particle-packed columns are the most used. Recently, monolithic columns have been established as promising alternative materials for nano-LC. A monolith is defined as a single piece of a porous material, and they are usually prepared by in situ polymerization. These materials are widely used in nano-LC due to their advantages of easy preparation, high permeability, fast mass transfer, low separation time, and simple production [11]. The most popular monolithic stationary phases are silica or organic polymeric ones. The main advantage of this type of column is that it has a highly porous structure and thereby ensures low resistance to mobile phases and low back pressure [15]. The use of monolithic columns as a stationary phase may provide a large surface area with a great opportunity for advanced and highly sensitive food analyses by nano-LC [8]. Lauryl methacrylate (LMA)-based monolithic columns are highly hydrophobic materials which are suitable for the chromatographic separation of hydrophobic compounds such as CoQ10. Proteins and some glycoproteins were separated using a boronate affinity monolith (e.g., a 100 µm i.d.) via cation exchange and hydrophobic interactions by nano-LCs [16]. 

Therefore, the main objective of this study is to develop a new LMA-based monolith to determine the amount of CoQ10 in pistachio samples, which leads to an investigation into the potential use of pistachio samples as natural sources of CoQ10. These samples could also be used for the development of new food products, providing enough amount of CoQ10 to be a natural supplement for the human body. The specific objectives were: (i) to develop a new monolith with a 50 µm i.d. for use in nano-LC; (ii) to develop a new analytical assay for CoQ10 analysis in food samples; and (iii) to determine the content of CoQ10 in pistachio samples.

## 2. Results and Discussion

### 2.1. Preparation and Characterization of the Monolithic Column with 50 µm i.d.

It is of utmost importance to achieve a high separation efficiency in any type of monolithic column in a capillary/nano-LC [17]. In this study, the preparation of a new monolithic column with a narrow capillary i.d. (e.g., 50 µm) was attempted using LMA as the main monomer and EDMA as the crosslinker by in situ polymerization. The polymerization solution content for the preparation of a new monolithic column described in this study was optimized according to the paper reported by Aydoğan et al. [16]. In that study, a cation exchange–hydrophobic affinity monolithic column with a 100 µm i.d. was prepared and applied to small molecule and protein separation. In this study, LMA was used as the main monomer for the preparation of the monolithic column. Various contents of both monomers and porogenic content (e.g., the amount of PEG in MeOH) were also investigated. In this optimization, when the increased content of LMA (more than 24% (*w/w*)) was attempted, it was shown that no flow rate could be obtained due to the blockage of the monolithic column. In light of the preliminary results, LMA (23.62 wt%) and EDMA (11.65 wt%) were used for the preparation of the polymeric solution, while the PEG content in the MeOH was used at 64.5 wt%. 

A higher content of LMA affected the structure of stationary phase. This information was obtained by the SEM images taken of different segments in the column. SEM images of the column with a 50 µm i.d. are given in Figure 1A,B. A well distribution to the inner surface of the capillary, Figure 1A, with homogenous structure, Figure 1B, can be observed.

These results indicated that nanoglobules up to 300 nm could be obtained. The mechanical stability of the monolithic column was evaluated by measuring the column back pressure using ACN:H_2_O (80:20%, *v/v*) at various flow rates. Figure 2A,B exhibit the mechanical stability of the monolithic column with different mobile phases, which shows a good relationship in both mobile phases (e.g., more than 0.9994). 

Three monolithic columns were prepared under the same chemical conditions. These columns were tested for the run-to-run, day-to-day, and column-to-column reproducibility in nano LC, which were assessed by the relative standard deviations (RSDs). Run-to-run, day-to-day, and column-to-column reproducibility values were found at approximately 2.1%, 3.2%, and 3.1% (*n* = 3), respectively. A reproducible retention with ethylbenzene as the test probe (thiourea as void marker) was shown as ≤3.18% of the RSD values. This was obtained using the mobile phase of ACN/H_2_O 85/15% (*v/v*), while no change in retention time was shown, which takes less than 1.8 min at a flow rate of 800 nL/min (see Figure 3). Different flow rates of up to 3000 nL/min were applied, while no changeable peak shape was shown. The results indicated that a promising hydrophobic monolith could be achieved. 

The chromatographic characterization of the monolithic column was performed using five homologous ABs including toluene, ethylbenzene, propylbenzene, butylbenzene, and pentylbenzene. The isocratic separation was attempted using ACN:H_2_O (80:20%, *v/v*) as a mobile phase at a flow rate of 600 nL/min. The conditions were optimized according to the published article [18,19]. Figure 4A shows the chromatographic separation of ABs with the monolith. As shown here, no sufficient retention could be achieved. When the ACN content was decreased to 60%, the separation was improved and the baseline separation could be obtained (see Figure 4B), which shows that hydrophobic interactions play an essential role in separating the ABs. These results can be attributed to the hydrophobicity of the column originating from LMA. 

The efficiency of the developed LMA-based column was assessed by Van Deemeter-plots for ethylbenzene, which is a composite curve that is governed by three terms which are largely controlled by the stationary phase particle size (*d*p), and the diffusion coefficients (*D*m) of the solute [20].

The monolithic column yielded 9 µm over at the linear mobile phase velocity of 1 mm/s, corresponding to flow rate of 500 nL/min (see Figure 5), which was produced over N~35,000 plates/m.

### 2.2. Study Design

Animal- and plant-based foods include CoQ10, which is part of our diet. Figure 6 shows the structure of CoQ10, which comprises a quinone group and a hydrophobic side chain of isoprenoid units. 

The amount of CoQ10 in foods ranges from 1 to 10 mg/kg [2]. One of the richest dietary sources of CoQ10 are nuts, such as pistachios, which can provide up to 50 mg/kg. The body of an average, healthy adult contains up to 1.5 g in relation to CoQ10. Several factors may affect the levels of CoQ10 [21], and several diseases can occur in the absence of CoQ10. In this sense, the development of new and advanced techniques for the analysis of CoQ10 is an essential issue. Various LC-UV systems were used for the analysis of CoQ10 in different samples such as meat [9], bee pollen [10], and several other food resources [22]. In our recent studies, various monolithic columns were prepared and used for the analysis of several anions in honey [23], CAP and CAPG in milk and honey [12], casein variants in milk [13]. In this study, a new monolithic column with a 50 µm i.d. was prepared for the analysis of CoQ10 in pistachio samples. The prepared column, containing a hydrophobic stationary phase, is suitable for the chromatographic separation of CoQ10, which is based on hydrophobic interactions. The developed column could be well-separated via a hydrophobic interaction mechanism, leading to the retention of CoQ10. 

### 2.3. Loading Capacity

Column loading capacity can intensely affect peak shape because the surface area of the column plays a crucial role. The capacity depends on the inner diameter of the column. The sample loading on the separation of CoQ10 was investigated by injecting the various concentrations of CoQ10, ranging from 0.05 to 5 mg/mL, while the injected sample volume of 20 nL was applied. The loading capacity for CoQ10 was found to be 0.4 mg/mL. It was obtained using the corresponding peak width at half-height (W_1/2_) while increasing by 10% over the peak width at a small sample volume. Figure 7 shows the test results of a 15 cm, LMA-based monolith. As is shown here, the W_1/2_ for CoQ10 at 0.4 mg/mL increased by 25% over the peak width. These results exhibited a good loading capacity of the monolith for the separation of CoQ10. 

### 2.4. The Optimization of Sample Preparation and Chromatography Conditions for Nano-LC

In order to optimize the conditions, several parameters, such as the mobile phase content, flow rate, and injection volume, were investigated, while some others were replaced due to the use of hydrophobic compounds, according to published literature. In this sense, CoQ10 could be well dissolved in several solvent systems such as can:2-PrOH or MeOH:2-PrOH, while EtOH:2-PrOH is not suitable in nano-LC because of the low flow rate applied (e.g., 500 nL/min). Considering the published literature [9,10], the solvent system with MeOH:2-PrOH:EtOH was tested at various ratios. The ratio of each solvent was then selected as MeOH:2-PrOH:EtOH (2:1:1%, *v/v*), respectively. An ultrasound sonicator (BANDELIN, KE76 probe) homogenizator was used for the sample preparation in order to obtain homogeneous samples (see Appendix A). It is noteworthy that the use of the ultrasound tip sonicator for the sample preparation allowed for a better homogeneous solution, leading to the good analysis of CoQ10 in the pistachio samples. As is shown in Appendix A, the samples could be successfully prepared before their injection to the nano-LC system. As a hydrophobic agent in the structure of the monolithic column, LMA provided hydrophobic interactions. Therefore, the monolith strongly interacted with CoQ10, allowing for its retention under neutral conditions. Various flow rates, ranging from 300 nL/min to 800 nL/min, were applied for the separation of the CoQ10 in nano-LC. This was evaluated with the one-by-one injection of the CoQ10 standard. The chromatographic separation conditions could be selected with the prepared monolithic nano-column using the mobile phase containing 2:1:1% (*v/v*) MeOH/EtOH/2-PrOH at a flow rate of 500 nL/min. The nano-LC-UV chromatograms demonstrated maximum absorption values at around 275 nm. Although various detection wavelengths, such as 210 and 278, were attempted, a stable baseline for CoQ10 was found at 275 nm, which was consistent with the literature [24]. The kinetic curves obtained in the presence of different CoQ10 concentrations (see Figure 8) showed a good retention of CoQ10 on the monolithic column. 

### 2.5. Method Validation

The method validation with LOD, LOQ and linearity as well as sensitivity using the pistachio samples were attempted using nano-LC with the developed monolith. In order to evaluate the repeatability, blank pistachio samples were spiked at concentration levels of 0.02–2000 µg/kg and injected five times each (*n* = 5) over three consecutive days. Table 1 displays the RSD values of CoQ10. It can be shown that the values changed in the range between 1.5 and 1.9 for the same day and for different days, respectively. 

The spiked pistachio samples at the three concentration levels were used for the measurement of recovery and repeatability, while the calibration curves could be obtained using the CoQ10 standard. All values, including LOD and correlation coefficients (R^2^), are given in Table 1. The recovery values were calculated using the literature [12], and are based on the ratio of the mean peak area of an analyte spiked before extraction to the mean peak area of an analyte spiked post-extraction, multiplied by 100 by carrying out five replicates at each spiked level (*n* = 5), while the results exhibited a good linearity (R^2^ > 0.99). 

Figure 9A–D show representative chromatograms for the separation of CoQ10 in (A) the blank sample and (B–D) the matrix-matched standards with different concentrations of CoQ10, demonstrating that CoQ10 could be sensitively determined in the samples. 

The LOD and LOQ values were 0.05 and 0.48 µg/kg, respectively. Various LOD and LOQ values for CoQ18 were reported in the literature. For instance, the LOD and LOQ values were found to be 0.16 and 0.35 mg/kg, respectively, determined using HPLC with a 4.6 mm i.d. column [10]. It could be seen that the use of a 50 µm i.d., reversed-phase column led to the sensitive detection of CoQ10. Finally, accuracy and precision were examined using spiking pistachio samples, leading to a good accuracy in the range of 85–105%. 

### 2.6. CoQ10 Analysis in Pistachio Samples

The developed method was applied to pistachio samples obtained from the Pistachio Research Institute in Türkiye. In total, six pistachio samples (see Appendix A) were analyzed. Table 2 shows the CoQ10 content of six pistachio samples with a variation in the contents amongst the samples. The highest amount for CoQ10 was found in Tekin, which was around 42.16 µg/g, while the other samples showed a close content of CoQ10. 

## 3. Materials and Methods

### 3.1. Chemicals

All chemicals were of analytical reagent grade. Ethylenedimethacrylate (EDMA, 99%), acetonitrile (LC-MS grade), and 3-trimethoxysilylpropyl methacrylate (TMSPM, 98%) were purchased from Sigma-Aldrich. 2.2′-Azobisizobutyronitrile (AIBN) and methanol (LC-MS grade) were purchased from Merck A.G (Darmstadt, Germany). A reducing union (1/16′ to 360 μm) SS, for FS tubing (S/N: L-P112240) from VICI Valco was used for the capillary connection between the monolithic column and the flow cell of detector. A TSP-050375-fused silica capillary with a 50 μm i.d. and a 363 μm o.d. (Lot: BUHT02A) was obtained from BGB Analytik. An amount of 2- propanol was purchased from Carlo Erba (for HPLC, isocratic grade). Ethanol (LC grade) was purchased from Isolab. Lauryl methacrylate (LMA, Lot: MKCL9070) and poly (ethyleneglycol) (PEG, Lot: BCCD8009) were supplied from Sigma-Aldrich. Coenzyme Q10 (≥98, HPLC, Lot: SLCF9505) was purchased from Sigma-Aldrich, and sodium chloride (NaCI) was supplied from Merck.

### 3.2. Instruments

A Dionex Ultimate 3000 Series ProFlow nano LC system from Thermo Scientific was used. The system contains the following parts: a solvent rack (SRD-3400), NCP-3200RS pump, a variable wavelength UV-Vis detector, −3400RS with a flow cell of 3 nL, and an autosampler with WPS-3000TPL RS. An ultrapure water system (Direct-Q^R^-3) from Milipore Corporation (Billerica, MA, USA) was used for obtaining pure water. This ultrapure water system also included an LC pack solution part (LC-Pak^R^ Polisher Catalogue No LCPAK0001), which provides ultrapure water for nano-LC and stock solutions. A WISD ultrasonic bath was used for obtaining a homogeneous polymerization solution for the preparation of the monolithic column. A BUTCH Rotavapor (R-300) was used for the evaporation process. A HERMLE Z327 K was used for centrifugation. An ultrasound sonicator (BANDELIN, KE76 probe) homogenizator was utilized to obtain a homogeneous mixture for the pistachio samples.

### 3.3. The Preparation of Monolithic Column

A silanization procedure was applied to a fused silica capillary column according to a previous study [18]. Briefly, a fused silica capillary with a length of 15 cm was silanized using 50% (*v/v*) 3-(trimethoxysilyl)propyl methacrylate in methanol, plugged with a GC septa, and reacted with at 35 °C for 24 h. The silanized capillary column was washed with MeOH to remove unreacted material. In order to prepare the monolith, the reactants were weighed into an Eppendorf tube with 2 mL of a 1.1 g total polymeric solution comprising LMA (23.62%, *w/w*) as the main monomer, EDMA as the crosslinker (11.65%, *w/w*), AIBN (0.03%, *w/w*) as the thermal initiator, and the porogenic solvents with PEG-methanol solution (64.7%, *w/w*). This solution was mixed to obtain the polymerization solution. The final polymerization solution was then injected into the silanized, fused silica capillary column using a syringe. The column was plugged at both ends with a septa, and the polymerization of the monolithic column was then performed at 65 °C for 20 h in a water bath. After polymerization, the resulting column was washed with ACN:H_2_O (80:20%, *v/v*) at a flow rate of 600 nL/min. 

### 3.4. Chromatographic Conditions

Alkylbenzenes (ABs) were dissolved in ACN and H_2_O. The sample solutions were freshly prepared and stored at +4 °C. The nano-LC detection wavelength was set at 200 nm for ABs and 275 nm for the analysis of CoQ10. The mobile phase used for the chromatographic separation of CoQ10 was methanol, 2- propanol, and ethanol (2:1:1 *v/v*). The flow rate was set at 600 nL/min and the injection volume was 20 nL.

### 3.5. Extraction of Coenzyme Q10 from Pistachio Samples

CoQ10 was extracted from six different pistachio samples using a previously reported solvent extraction method [25]. Briefly, a 0.5 g sample was weighed and homogenized in 10 mL of the organic solvent. This was sonicated and added to the ultra-turrax high speed homogenizer at 9500 rpm for 5 min. An amount of 5 mL of the homogenized sample was placed into an extraction tube, mixed with 2 mL of ethanol, and centrifuged at 8000 rpm for 2 min, after which the solid residue was discarded. Then, after a vigorous mixture, the tube was quickly centrifuged at 5000 rpm for 10 min. The organic layer was transferred and the lower layer was re-extracted twice with 5 mL of ethanol. The mixture of the layers was evaporated using a rotary evaporator and the dried residue was dissolved in ethanol, 2- propanol, methanol (2:1:1%, *v/v*). 

### 3.6. Method Validation

A method validation with LOD, LOQ, and linearity was attempted to evaluate the developed nano-LC method with the LMA-based monolithic column. This procedure was performed according to published literature [12,23]. The standard solutions of CoQ10 were prepared using a stock solution with 200 mg/kg, which was stored at 4 °C in the refrigerator. Calibration standards for the CoQ10 were prepared with the concentrations of 0.01, 0.1, 1, 50, 100, and 200 mg/kg using stock solutions. These solutions were prepared using a solution of ethanol, 2- propanol, methanol (2:1:1 *v/v*). The concentration of the standards was plotted to the respective internal standard (x) versus the peak area ratio (y). The LOD (*n* = 3) and LOQ (*n* = 5) were calculated using the spiked samples [12]. Recovery and repeatability values were calculated using spiked samples at three concentration levels. 

## 4. Conclusions

A new, LMA-based, monolithic nano-column with a 50 µm i.d. was developed for nano-LC. After characterization, the final monolithic column was applied for the analysis of CoQ10 in six pistachio samples. The developed method could be successfully validated in terms of repeatability, recovery, linear range, and sensitivity using six pistachio samples. All the results showed that the proposed technique is reliable for the rapid and sensitive separation and quantification of CoQ10 in samples. The utilization of the developed monolithic column can also be applied for various food samples. The new materials (e.g., the monolithic column) are promising tools for advanced food analysis and allow for the development of new techniques. The developed method using nano-LC provides many advantages over the classical methods.

## Figures and Tables

**Figure 1 molecules-28-01423-f001:**
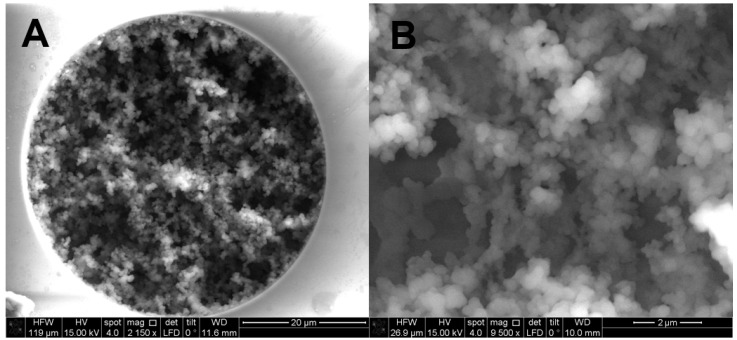
SEM images of the monolithic nano-column with magnification (**A**) ×2150 (**B**) ×9500.

**Figure 2 molecules-28-01423-f002:**
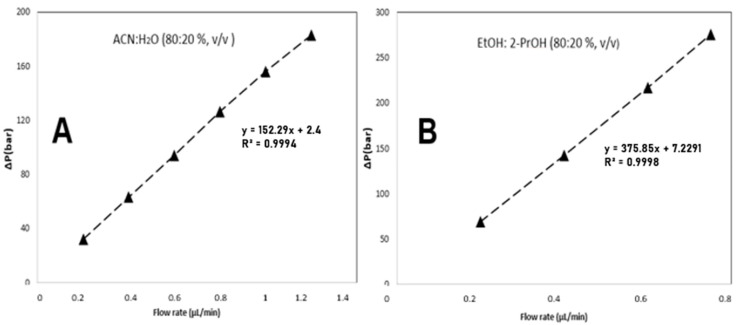
Plot of the monolithic column’s back pressure versus flow rate with the mobile phases (**A**) ACN/H_2_O (80:20%, *v/v*) and (**B**) EtOH/2-PrOH (80:20%, *v/v*).

**Figure 3 molecules-28-01423-f003:**
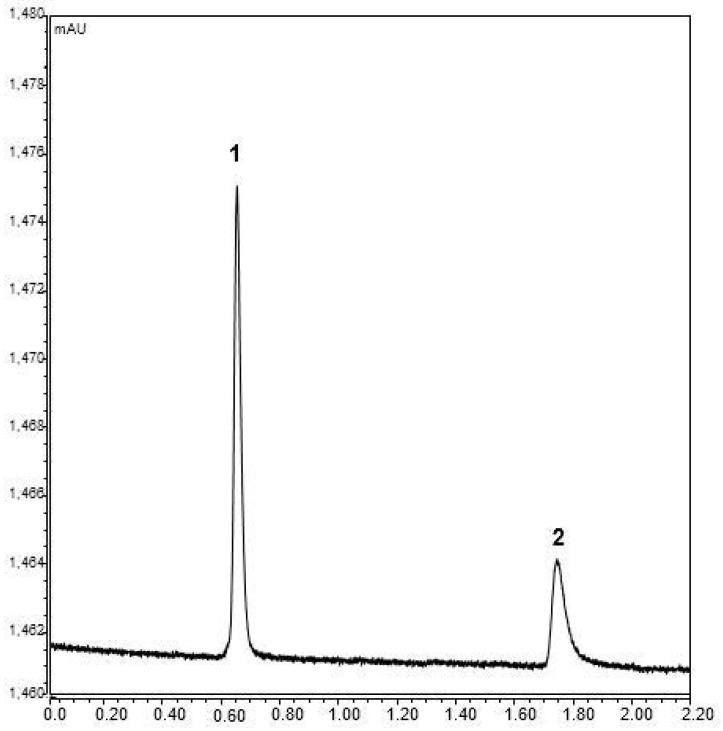
The separation chromatogram of the text mixture used for performance evaluation. Chromatographic conditions: mobile phase, 85%ACN:15% H_2_O; flow rate, 800 nL/min; injection volume: 4 nL; detection wavelength: 254 nm; peaks: (1) thiourea and (2) ethylbenzene.

**Figure 4 molecules-28-01423-f004:**
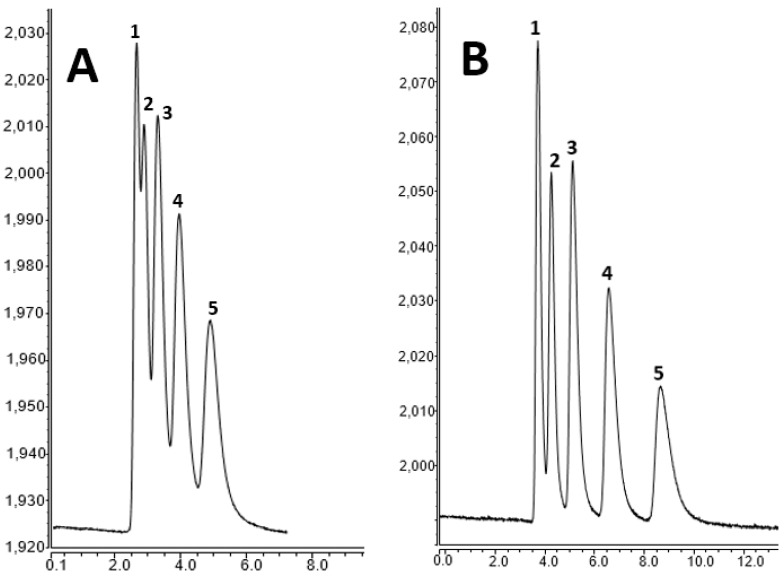
Chromatographic separation of alkylbenzenes on the monolithic nano-column at different ACN contents. Chromatographic conditions: (**A**) mobile phase: 80%ACN:20%H_2_O (*v/v);* analyte concentration: 0.04 µg/mL; detection wavelength: 200 nm; flow rate: 600 nL/min; order of peaks: (1) toluene (2) ethylbenzene, (3) propylbenzene, (4) butylbenzene, and (5) pentylbenzene. (**B**) Mobile phase: 60%ACN:40%H_2_O (*v/v);* the other conditions are same as (**A**).

**Figure 5 molecules-28-01423-f005:**
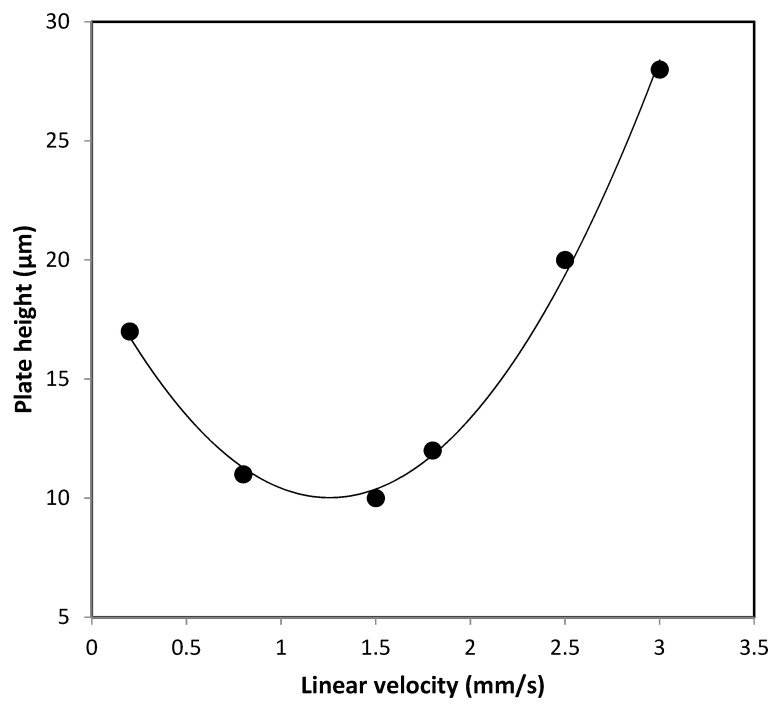
Van Deemeter curves of the monolith. Chromatographic conditions: mobile phase, 85%ACN:15% H_2_O, injection volume: 4 nL.

**Figure 6 molecules-28-01423-f006:**
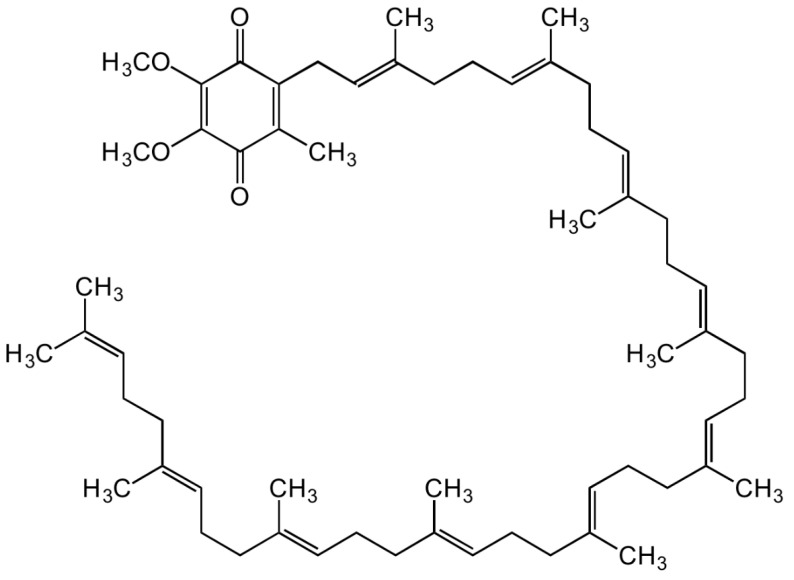
The structure of CoQ10.

**Figure 7 molecules-28-01423-f007:**
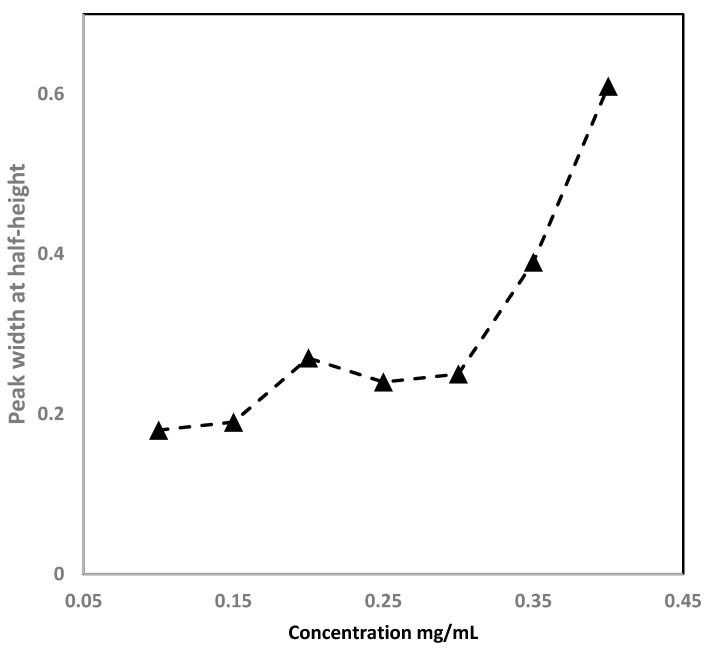
The loadability tests of the 15 cm long monolith using CoQ10. Chromatographic conditions: mobile phase: methanol, 2- propanol, ethanol (2:1:1 *v/v*) with 0.1% TFA. The flow rate was set at 500 nL/min and the injection volume was 20 nL.

**Figure 8 molecules-28-01423-f008:**
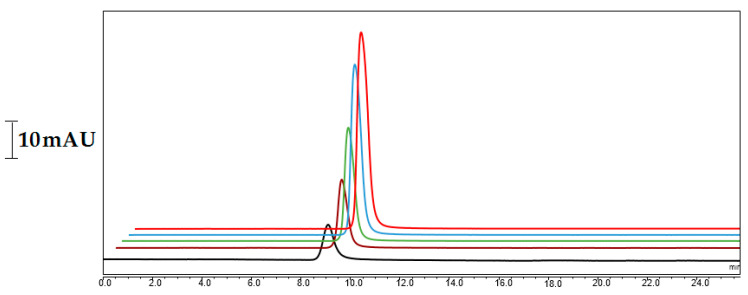
Kinetic curves obtained in the presence of different CoQ10 concentrations (e.g., 0.5 µg/mL to 5000 µg/mL).

**Figure 9 molecules-28-01423-f009:**
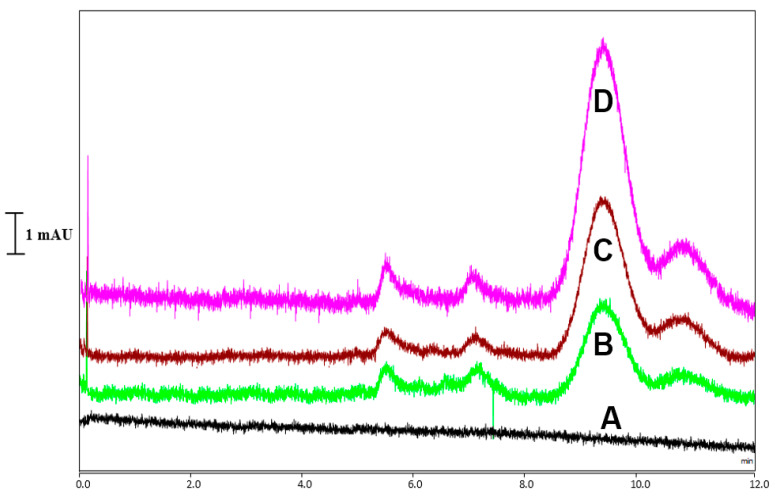
Separation chromatogram of CoQ10 in the A blank and B–D matrix-matched standards with different concentrations of CoQ10.

**Table 1 molecules-28-01423-t001:** Validation parameters for CoQ10 determination in nano-LC.

Analyte	LOD µg/kg	LOQ µg/kg	R^2^	Precision (RSD%) ^a^
				Inter day Intra day
CoQ10	0.05	0.48	0.9992	1.5 1.9

LOD: Limit of detection; LOQ: Limit of quantification; ^a^ RSD (*n* = 5).

**Table 2 molecules-28-01423-t002:** Contents of CoQ10 in a Pistachio samples.

Samples	Barak Yıldızı	Tekin	Halebi	Uzun	Siirt	Kırmızı
CoQ10 (µg/g)	30.5 ± 0.4	42.16 ± 0.6	17.21 ± 0.6	10.21 ± 0.4	28.21 ± 0.5	30.2 ± 0.8

## Data Availability

The data that support the findings of this study are available from the corresponding author upon reasonable request.

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
