# Peer review of "Nano-Liquid Chromatography with a New Monolithic Column for the Analysis of Coenzyme Q10 in Pistachio Samples"

_molecules, 2023, doi:10.3390/molecules28031423_

Round 1

Reviewer 1 Report

Dear Authors,

Work is interesting.

Followin suggesrltions

Abstract'- restructur...is it about column or coenzimeq10

Conclusion must be more clear

Introduction- pleas explain why pistachio samples

Also chemical characterisation relevant to colmn developement is missing

Materials:

Column preparation should be more clear...how particle size were determined...sem scanning how it was performed....in situ polimerisation is there experimental designe applied...if not on which study polymerisation is based

Line 114 rines how

Line 121 why this mobile phase

Why that concentration range for calibration was choosen

Line 144 what concentration levels what points were choosen for that

Results

Where is evidence that particles were 50um

Sem is blurred

Pleas add in supplementary screenshot of chroatograph pictures (connected to figure 3)

Loading capacity is not fully explained

Inter day and intra day precission is not explained in materials

Rsd ussualy is don with n10 why n5 explain

Line 289 what is relevant institute it is not mentioned in methods please add it

Improve results to support conclusion

Best of luck

Author Response

Dear Authors,

Work is interesting.

The authors are thankful for reviewer’s positive assessment of the manuscript. The original manuscript was revised according to reviewer comments.

Followin suggestions

Abstract'- restructur...is it about column or coenzimeq10

A new monolithic column was developed for the analysis of CoQ10 in the samples.

Conclusion must be more clear

This section was revised as suggested.

Introduction- pleas explain why pistachio samples

It was revised as suggested.

Also chemical characterisation relevant to colmn developement is missing

It was performed using SEM as well as chromatographic analyses.

Materials:

Column preparation should be more clear...how particle size were determined...sem scanning how it was performed....in situ polimerisation is there experimental designe applied...if not on which study polymerisation is based.

Pore size is important but not critical for this column. SEM images was taken by cutting of monolith with 3 cm length and introduced into the sysytem.

Line 114 rines how

It was revised as suggested.

Line 121 why this mobile phase

CoQ10 could be well dissolved in that mobile phase.

Why that concentration range for calibration was choosen

It was given section 2.6.

Line 144 what concentration levels what points were choosen for that

Considering the content of CoQ10 in the samples according to literature, those levels were attempted.

Results

Where is evidence that particles were 50um

We apologize for this. The column i.d. was 50 um.

Sem is blurred.

Although we have several others, the better ones were put on the article.

Pleas add in supplementary screenshot of chroatograph pictures (connected to figure 3).

The related data was replaced and corrected.

Loading capacity is not fully explained

The related part was rewritten.

Inter day and intra day precission is not explained in materials

It was given at last part of section 3.1.

Rsd ussualy is don with n10 why n5 explain

It was 5, which means replicates number

Line 289 what is relevant institute it is not mentioned in methods please add it

 It was revised as suggested.

Improve results to support conclusion

Best of luck

Thank you

Reviewer 2 Report

Within the paper entitled ''Nano-liquid chromatography with a new monolithic nano-column for the analysis of Coenzyme Q10 in Pistachio samples'' the authors, Aydogan et al., have presented detailed analytical chemistry approaches for studying the coenzyme Q10 in the food (in this case the Pistachio) samples. I really appreciated the wide range of the topics touched within this work (very clearly presented within this study, what does not happen so often). Food-related science is a rapidly developing discipline of knowledge in the current times. That is why more and more methods as well as the analytical techniques are being employed to monitor, and to eventually solve this problem. Metabolomics analyses performed with NMR and Mass Spec techniques are one of the best approaches for that. In order to improve this paper authors should refer to the test placed below:   There is a large potential in this area of research (especially metabolomics), because in the recent years a lot of the developments have been observed in the analytical as well as spectroscopic methods of the identification of the compounds (natural ones as well as pollutants that can be present in fruits or vegetables) in a variety of types of different tissues (Molecules. 2020 Oct 9;25(20):4597. doi: 10.3390/molecules25204597). The NMR spectroscopy techniques offered recently the next step in more detailed studying the issue related to the new potentially active or potentially toxic compounds in different types of the bio-material (like a tissue of an animal or plant) on the molecular level, one can look into direction of metabolomics, targeted and untargeted analyses, and its wide possibilities that would complement the complementary studies of the plant/animal material. Thanks to this approach more useful markers can be identified. Moreover, it is worth mentioning at this point the recently reported developments in the methodology that is used in e. g. NMR-based metabolomics studies in the sensitivity as well as in the resolution of spectra (Mol Omics. 2021 Oct 11;17(5):719-724. doi: 10.1039/d1mo00118c). These improvements give new possibilities for a more comprehensive approach to the presented problems connected with the economically important aspects in medicine.   Moreover, there is still some more room for an improvement of this decent work of a food-related analytical chemistry on the active and biochemically useful compounds in foods we eat and the environment they come from. In order to further improve the quality of this study authors should refer to newer works studying and refer to the other findings in the different types of foods. The recently published work mentioned under the given citation should be quite helpful in doing this:   https://www.mdpi.com/2304-8158/10/6/1249 - very comprehensive review in food metabolomics that should be referred to within a small succinct paragraph. Besides this what i advice authors of the paper to ellaborate more on the statistical analyses they applied to their data and to use other tests (e. g. Akaike criterion) to verify the correlations shown on graphs they already got.

And to use other tests (e. g. Akaike criterion) to verify the correlations shown on graphs they already got.

After implementation of these changes mentioned above will surely make this work more attractive to the broader group of readers.

Author Response

Reviewer # 2

-Within the paper entitled ''Nano-liquid chromatography with a new monolithic nano-column for the analysis of Coenzyme Q10 in Pistachio samples'' the authors, Aydogan et al., have presented detailed analytical chemistry approaches for studying the coenzyme Q10 in the food (in this case the Pistachio) samples. I really appreciated the wide range of the topics touched within this work (very clearly presented within this study, what does not happen so often).

The authors are thankful for reviewer’s positive assessment of the manuscript. The original manuscript was revised according to reviewer comments.

Food-related science is a rapidly developing discipline of knowledge in the current times. That is why more and more methods as well as the analytical techniques are being employed to monitor, and to eventually solve this problem. Metabolomics analyses performed with NMR and Mass Spec techniques are one of the best approaches for that. In order to improve this paper authors should refer to the test placed below:  

There is a large potential in this area of research (especially metabolomics), because in the recent years a lot of the developments have been observed in the analytical as well as spectroscopic methods of the identification of the compounds (natural ones as well as pollutants that can be present in fruits or vegetables) in a variety of types of different tissues (Molecules. 2020 Oct 9;25(20):4597. doi: 10.3390/molecules25204597). The NMR spectroscopy techniques offered recently the next step in more detailed studying the issue related to the new potentially active or potentially toxic compounds in different types of the bio-material (like a tissue of an animal or plant) on the molecular level, one can look into direction of metabolomics, targeted and untargeted analyses, and its wide possibilities that would complement the complementary studies of the plant/animal material. Thanks to this approach more useful markers can be identified. Moreover, it is worth mentioning at this point the recently reported developments in the methodology that is used in e. g. NMR-based metabolomics studies in the sensitivity as well as in the resolution of spectra (Mol Omics. 2021 Oct 11;17(5):719-724. doi: 10.1039/d1mo00118c). These improvements give new possibilities for a more comprehensive approach to the presented problems connected with the economically important aspects in medicine.   Moreover, there is still some more room for an improvement of this decent work of a food-related analytical chemistry on the active and biochemically useful compounds in foods we eat and the environment they come from. In order to further improve the quality of this study authors should refer to newer works studying and refer to the other findings in the different types of foods. The recently published work mentioned under the given citation should be quite helpful in doing this:   https://www.mdpi.com/2304-8158/10/6/1249 - very comprehensive review in food metabolomics that should be referred to within a small succinct paragraph. Besides this what i advice authors of the paper to ellaborate more on the statistical analyses they applied to their data and to use other tests (e. g. Akaike criterion) to verify the correlations shown on graphs they already got.
After implementation of these changes mentioned above will surely make this work more attractive to the broader group of readers.

It was revised as suggested kindly see Introduction section

Reviewer 3 Report

The authors have developed a new nano-column and demonstrated its effectiveness at extracting CoQ10 from different pistachio samples.

The research appears significant enough to warrant publication. However, significant editing of the language and spelling will greatly increase readability. I include a few minor comments that I believe would increase readability.

Further comments.

1)      Please reference the method for collecting SEM data, I could not find it in the paper. In addition, Fig 1 should be improved by cutting off the data in the bottom of the field and including a readable well-sized scale bar in the image itself.

2)       The methods section is otherwise reasonably complete. Please use either 50:25:25 or 2:1:1 consistently (I think you want to use percentages).

3)      Figure 3, one of those sides is not 80:20, I think the left side is 60:40?

4)      Line 223 2.1 (units) ID column? I assume you mean mm but please include it.

5)      Figure 5 – what concentrations are shown here?

6)      Table 1 – I prefer to see a graph that shows LOD and LOQ instead of seeing them in a table. Especially when LOD is 10x smaller than LOQ. I understand there is a good R2, but even so.

7)      Figure 6 – what concentrations are shown here?

Author Response

Reviewer # 3

-The authors have developed a new nano-column and demonstrated its effectiveness at extracting CoQ10 from different pistachio samples. The research appears significant enough to warrant publication. However, significant editing of the language and spelling will greatly increase readability. I include a few minor comments that I believe would increase readability.

 The authors are thankful for reviewer’s positive assessment of the manuscript.

Further comments.

1)      Please reference the method for collecting SEM data, I could not find it in the paper. In addition, Fig 1 should be improved by cutting off the data in the bottom of the field and including a readable well-sized scale bar in the image itself.

      SEM images was taken by UNAM (Ankara-Turkey). We put them as it is. The information can be seen on it.

2)       The methods section is otherwise reasonably complete. Please use either 50:25:25 or 2:1:1 consistently (I think you want to use percentages).

It was revised as suggested.

3)      Figure 3, one of those sides is not 80:20, I think the left side is 60:40?

It was corrected.

4)      Line 223 2.1 (units) ID column? I assume you mean mm but please include it.

  use percentages).

         It was revised.

5)      Figure 5 – what concentrations are shown here?

           It was given in the revised text.

6)      Table 1 – I prefer to see a graph that shows LOD and LOQ instead of seeing them in a table. Especially when LOD is 10x smaller than LOQ. I understand there is a good R2, but even so.

Thank you. Table was prepared in three-lines format.

7)      Figure 6 – what concentrations are shown here?

It was given in the revised text.

Reviewer 4 Report

The authors synthesized the hydrophobic monolithic capillary for nano-HPLC application. This is demonstrated for quantitative analysis of Coenzyme Q10 in the Pistachio extracts. It is always interesting for me to see a wider range of applications with the monolith technology. However, this work requires significant improvement especially with the English polishing, experimental details and the additional results. The main concerns are listed below.

-Information about sample coating for SEM, SEM experimental conditions, capillary pre-treatment and how to silanize the capillary is missing.

-Please clearly inform how to prepare the reagents with w/w composition. Did the author weigh the overall solution before or after adding the porogenic solvent? 

-Lines 169-170: … measuring the column back pressure using ACN:H2O (80:20 %, v/v) at various flow rates … A good reason should be provided why the authors investigated this using ACN:H2O and EtOH:2-PrOH that are not the mobile phase composition applied in the separation.

-Instead of showing the pressure vs flow plots in Figure 2, the van Deemter curves of significantly retained compounds should be provided. The van Deemter curves are strictly required in this study since the authors claimed at Lines 245-250 that “Various flow rates ranged 300nL/min to 800 nL/min were applied for the separation … The chromatographic separation conditions could be optimized … using the mobile phase containing 50/25/25% (v/v) MeOH/EtOH/2-PrOH at a flow rate of 500 nL/min. T”. This indicates the flow rate of 500 nL/min should result in the minimum plate heights in the van Deemter curves. Please provide this evidence.

-Figure 3B caption is incorrect, please correct the mobile phase composition à 60:40?.

- Lines 180-183: ”A reproducible retention with ethylbenzene as the test compound (thiourea as void marker) was shown as ≤3.18 % of RSD values … using the mobile phase of ACN/H2O 80/20 % ... no change in retention time was shown.” Based on the chromatograms in Figure 3, it could be that toluene and ethylbenzene coeluted closely to the void time. Thus, these two compounds with very weak retentions are not suitable for the repeated analyses. For example, the similar time of ethylbenzene in all the columns may be just that they all have the similar void time. The authors should select the compounds with the stronger retention such as pentyl benzene or better use the target analyte (CoQ10) in the reproducibility test. Otherwise, please provide the experimental evidence to address my concern here such as that with the raw chromatograms showing both retention times of ethylbenzene and the thiourea void marker.

-Line 226: “The loading capacity for CoQ10 were found as 0.4 mg/mL” The authors should inform how to calculate this with proper citation.

-Lines 227-228: ”This result exhibited the higher loading capacity for the developed monolith with 50 µm ID for CoQ10” This discussion could be against the evidence that the authors provided since the 1 µg or more for 2.1 ID column was mentioned (Line 223). However, the authors reported their capacity in the concentration unit of 0.4 mg/mL (Line 226). This is 400 ug/mL = 0.0004 ug/nL. If my quick check here is correct, the capacity of the authors' column is only 0.008 ug (with 20 nL injected sample volume). This is clearly “lower loading capacity for the developed monolith with 50 µm ID for CoQ10” NOT “higher”.

- It appears that all the compounds could not be completely eluted within 12 min (e.g. see the chromatogram C in Figure 6). Did the authors stop the run at 12 min? If yes, there should be the carryover leading to the false analyses in the next runs. If no, please provide the full chromatograms after 12 min.

-The authors did not provide related experimental results to claim the word “optimization” in section 3.4. This wording should be replaced with “selection” or "selected" throughout the manuscript. For example, Lines 236-237: Considering the published literature [5,6], the mobile phase was optimized as MeOH:2-PrOH:EtOH (50:25:25 %, v/v). à This is only following the literature (NOT optimization). Such mobile phase condition was also achieved for the other researchers' work using the other columns (NOT the authors column). The chromatograms showing the mobile phase/flow rate variation are missing. The measurable results to show that the sample preparation was optimized are also missing.

- The information of different concentrations of CoQ10 in Figure 6 is missing.

-The authors calculated LOD and LOQ following the method of [9] (Line 143). However, the authors compared the values with LOD and LOQ from [6] (Lines 282-285). These two references are from different research groups and they could calculate LOD and LOQ using different approach. Without the evidence that [6] using the same approximation approach, such comparison should be avoided or provided with careful notes.

- Lines 286-287: "Finally, accuracy and precision were examined using spiking pistachio samples, leading to good accuracy in the range of 85-105%." The method to calculate accuracy in this sentence should be informed. The precision values following this sentence are also missing.

My other concerns are listed below.

-Line 57: Please define the words of “green and advanced chromatography”.

-Line 77: Regarding “… (i) to develop a new monolith with 50 µm i.d. …”, please provide the introduction how 50 µm i.d. could be better than 100 µm i.d. mentioned in [14].

-Line 87: After reading most of the work, I believe the authors only presented the result for 50 µm id capillary confirmed by the size shown in the SEM image. Please correct the sentence “… 050375-fused silica capillary with 100 µm id … “.

-“2.3. The preparation of monolithic columns”. The column length information is missing. Please also indicate the applied column length in this section. It should also be informed in this section that 3 columns were synthesized for the repeatability test.

-In the section 2.4, ABs were dissolved in ACN and H2O. Could the authors provide the reason why this is not matched with the applied mobile phase “methanol, 2- propanol, ethanol”.

- Line 143: “LOD (n = 3) and LOQ (n = 5) were calculated by the spiked samples [9].” Please provide the detail about how to obtain the LOD LOQ (at which concentration ranges) and why were the replicates n values different?

-There is mismatch between the flow rates at Lines 121-122: “The flow rate was set at 600 nL/min” and Line 250: “500 nL/min”. Please explain why they were different?

-Please do not stretch the molecule picture in Figure 4.

-English and consistency should be carefully polished throughout the manuscript. The examples are provided below.

Lines 44-45: … due to food products include many ingredients …

Lines 57-58: … various type of columns …

Line 102: … ultrasonic bath was used for to obtain …

Line 222: “This capacity depends the innerdiameter of

Line 250 “absorbans

Lines 277 and 281: “concentations”

Line 295 “with is

id or i.d.

Author Response

Reviewer # 4

-The authors synthesized the hydrophobic monolithic capillary for nano-HPLC application. This is demonstrated for quantitative analysis of Coenzyme Q10 in the Pistachio extracts. It is always interesting for me to see a wider range of applications with the monolith technology. However, this work requires significant improvement especially with the English polishing, experimental details and the additional results. The main concerns are listed below.

The authors are thankful for reviewer’s positive assessment of the manuscript. The original manuscript was revised according to reviewer comments.

-Information about sample coating for SEM, SEM experimental conditions, capillary pre-treatment and how to silanize the capillary is missing.

More details were given in section 2.3

-Please clearly inform how to prepare the reagents with w/w composition. Did the author weigh the overall solution before or after adding the porogenic solvent? 

More details were given in section 2.3

-Lines 169-170: … measuring the column back pressure using ACN:H2O (80:20 %, v/v) at various flow rates … A good reason should be provided why the authors investigated this using ACN:H2O and EtOH:2-PrOH that are not the mobile phase composition applied in the separation.

We always use ACN:H2O for washing and measuring column back pressure of the prepared reversed-phase columns. It also shows column mechanical stability. In this case, EtOH:2-PrOH was also attempted due to tha fact that the mobile phase used was more viscose, which shows the suitability of the column.

-Instead of showing the pressure vs flow plots in Figure 2, the van Deemter curves of significantly retained compounds should be provided. The van Deemter curves are strictly required in this study since the authors claimed at Lines 245-250 that “Various flow rates ranged 300nL/min to 800 nL/min were applied for the separation … The chromatographic separation conditions could be optimized … using the mobile phase containing 50/25/25% (v/v) MeOH/EtOH/2-PrOH at a flow rate of 500 nL/min. T”. This indicates the flow rate of 500 nL/min should result in the minimum plate heights in the van Deemter curves. Please provide this evidence.

It was given in Fig 4. and following sententce was added to the revised text. “the column efficiency was assessed by Van Deemeter-plots for the compound, which is a composite curve that is governed by three terms, which are largely controlled by stationary phase particle size (dp), and diffusion coeffcients (Dm) of the solute”.

-Figure 3B caption is incorrect, please correct the mobile phase composition à 60:40?.

It was corrected.

- Lines 180-183: ”A reproducible retention with ethylbenzene as the test compound (thiourea as void marker) was shown as ≤3.18 % of RSD values … using the mobile phase of ACN/H2O 80/20 % ... no change in retention time was shown.” Based on the chromatograms in Figure 3, it could be that toluene and ethylbenzene coeluted closely to the void time. Thus, these two compounds with very weak retentions are not suitable for the repeated analyses. For example, the similar time of ethylbenzene in all the columns may be just that they all have the similar void time. The authors should select the compounds with the stronger retention such as pentyl benzene or better use the target analyte (CoQ10) in the reproducibility test. Otherwise, please provide the experimental evidence to address my concern here such as that with the raw chromatograms showing both retention times of ethylbenzene and the thiourea void marker.

The related data was checked and corrected, kindly see the last part of section 3.1

-Line 226: “The loading capacity for CoQ10 were found as 0.4 mg/mL” The authors should inform how to calculate this with proper citation.

It was revised as suggested.

-Lines 227-228: ”This result exhibited the higher loading capacity for the developed monolith with 50 µm ID for CoQ10” This discussion could be against the evidence that the authors provided since the 1 µg or more for 2.1 ID column was mentioned (Line 223). However, the authors reported their capacity in the concentration unit of 0.4 mg/mL (Line 226). This is 400 ug/mL = 0.0004 ug/nL. If my quick check here is correct, the capacity of the authors' column is only 0.008 ug (with 20 nL injected sample volume). This is clearly “lower loading capacity for the developed monolith with 50 µm ID for CoQ10” NOT “higher”.

Relevant part was revised and rewritten.

- It appears that all the compounds could not be completely eluted within 12 min (e.g. see the chromatogram C in Figure 6). Did the authors stop the run at 12 min? If yes, there should be the carryover leading to the false analyses in the next runs. If no, please provide the full chromatograms after 12 min.

There is no carryover. We did it at lower flow rate e.g. at 300 nL/min (see below Fig.), which means the retention could be obtained after optimization.  

-The authors did not provide related experimental results to claim the word “optimization” in section 3.4. This wording should be replaced with “selection” or "selected" throughout the manuscript. For example, Lines 236-237: Considering the published literature [5,6], the mobile phase was optimized as MeOH:2-PrOH:EtOH (50:25:25 %, v/v). à This is only following the literature (NOT optimization). Such mobile phase condition was also achieved for the other researchers' work using the other columns (NOT the authors column). The chromatograms showing the mobile phase/flow rate variation are missing. The measurable results to show that the sample preparation was optimized are also missing.

It means we try several solvent systems to dissolve CoQ10 but the solvents didnt solve properly, we also searched the literature and found the use of MeOH:2-PrOH:EtOH. The ingredients of each solvent were optimized. Kindly see section 3.4.

- The information of different concentrations of CoQ10 in Figure 6 is missing.

It was revised as suggested.

-The authors calculated LOD and LOQ following the method of [9] (Line 143). However, the authors compared the values with LOD and LOQ from [6] (Lines 282-285). These two references are from different research groups and they could calculate LOD and LOQ using different approach. Without the evidence that [6] using the same approximation approach, such comparison should be avoided or provided with careful notes.

In ref 6, the analysis of CoQ10 was performed using conventional HPLC (e.g. at 1ml/min)  but in this case we used nano-HPLC (e.g. 500nL/min). We think that the comparisons of the relevant results would be reasonable.

- Lines 286-287: "Finally, accuracy and precision were examined using spiking pistachio samples, leading to good accuracy in the range of 85-105%." The method to calculate accuracy in this sentence should be informed. The precision values following this sentence are also missing.

It was given in section 2.5. in the revised manuscript.

My other concerns are listed below.

-Line 57: Please define the words of “green and advanced chromatography”.

It means advanced technology

-Line 77: Regarding “… (i) to develop a new monolith with 50 µm i.d. …”, please provide the introduction how 50 µm i.d. could be better than 100 µm i.d. mentioned in [14].

The column with the lower column I.D. allows better sensitivity. 10.1002/jssc.200401902

-Line 87: After reading most of the work, I believe the authors only presented the result for 50 µm id capillary confirmed by the size shown in the SEM image. Please correct the sentence “… 050375-fused silica capillary with 100 µm id … “.

It was corrected.

-“2.3. The preparation of monolithic columns”. The column length information is missing. Please also indicate the applied column length in this section. It should also be informed in this section that 3 columns were synthesized for the repeatability test.

The relevant information was given in section 2.3.

-In the section 2.4, ABs were dissolved in ACN and H2O. Could the authors provide the reason why this is not matched with the applied mobile phase “methanol, 2- propanol, ethanol”.

ABs as the test probes were used for column characterization.

- Line 143: “LOD (n = 3) and LOQ (n = 5) were calculated by the spiked samples [9].” Please provide the detail about how to obtain the LOD LOQ (at which concentration ranges) and why were the replicates n values different?

More information was given in section 2.6.

-There is mismatch between the flow rates at Lines 121-122: “The flow rate was set at 600 nL/min” and Line 250: “500 nL/min”. Please explain why they were different?

The flow rate could be different according to the compound used.

-Please do not stretch the molecule picture in Figure 4.

It was revised as suggested.

-English and consistency should be carefully polished throughout the manuscript. The examples are provided below.

Lines 44-45: … due to food products include many ingredients …

Lines 57-58: … various type of columns …

Line 102: … ultrasonic bath was used for to obtain …

Line 222: “This capacity depends the innerdiameter of”

Line 250 “absorbans”

Lines 277 and 281: “concentations”

Line 295 “with is”

id or i.d.

All comments were revised as suggested. Also, Manuscript English was checked and corrected.

Round 2

Reviewer 1 Report

Dear Authors,

Minor spelling mistakes are still present. Recheck your  text. All corrections are done as requested. Figure 3 and 7 sshoild be drown from original softwer. They look painted, please correct.

Best of luck

Author Response

Dear Authors,

Minor spelling mistakes are still present. Recheck your  text. All corrections are done as requested. Figure 3 and 7 sshoild be drown from original softwer. They look painted, please correct.

Best of luck

Thank you for comment. The text was checked and corrected. The figures were just obtained from the software our ProFlow nano-LC system (e.g Chromline 7). 

Reviewer 4 Report

Most of my concerns have been well addressed. However, some issues below are still of concern requiring further improvement.

-The typical van Deemter curve of a monolithic column with good high permeability is expected to show lower C term with the flatter slope on the right hand side of the H vs v curve at the faster flow [R. Hayes et al. Journal of Chromatography A, 1357 (2014) 36–52]. The curve in Figure 4 did not represent that. Could this be related with the low permeability of the column in this study, e.g. caused inhomogeneous porosity, partial clogging? The discussion related to this should be provided.

-Even with what the authors provided as the improvement, the word "optimization/optimize" should be avoided. Otherwise, curve fitting, experimental design model should be provided in order to prove that the conditions that the authors were actually at the optimum points. “Selected/selection” could be the better.

-The authors did not properly answer the related question below.

Lines 180-183: ”A reproducible retention with ethylbenzene as the test compound (thiourea as void marker) was shown as ≤3.18 % of RSD values … using the mobile phase of ACN/H2O 80/20 % ... no change in retention time was shown.” Based on the chromatograms in Figure 3, it could be that toluene and ethylbenzene coeluted closely to the void time. Thus, these two compounds with very weak retentions are not suitable for the repeated analyses. For example, the similar time of ethylbenzene in all the columns may be just the results that they all have the similar void time. The authors should the compounds with the stronger retention such as pentyl benzene or better use the target analyte (CoQ10) in the reproducibility test. Otherwise, please provide the experimental evidence to address my concern here such as that with the raw chromatograms showing both retention times of ethylbenzene and thiourea void marker in all the cases.

At least, the %RSD results for the pentyl benzene or CoQ10 should be provided.

Author Response

Most of my concerns have been well addressed. However, some issues below are still of concern requiring further improvement.

The authors are thankful for reviewer’s positive assessment of the manuscript. The manuscript was revised according to reviewer comments.

-The typical van Deemter curve of a monolithic column with good high permeability is expected to show lower C term with the flatter slope on the right hand side of the H vs v curve at the faster flow [R. Hayes et al. Journal of Chromatography A, 1357 (2014) 36–52]. The curve in Figure 4 did not represent that. Could this be related with the low permeability of the column in this study, e.g. caused inhomogeneous porosity, partial clogging? The discussion related to this should be provided.

We examined the article and could say that the dispersion models between core-shell particles and monolithic columns are not same due to the fact that monoliths consist of a network of through pores separated by a polymeric skeleton.

-Even with what the authors provided as the improvement, the word "optimization/optimize" should be avoided. Otherwise, curve fitting, experimental design model should be provided in order to prove that the conditions that the authors were actually at the optimum points. “Selected/selection” could be the better.

It was revised as suggested.

-The authors did not properly answer the related question below.

Lines 180-183: ”A reproducible retention with ethylbenzene as the test compound (thiourea as void marker) was shown as ≤3.18 % of RSD values … using the mobile phase of ACN/H2O 80/20 % ... no change in retention time was shown.” Based on the chromatograms in Figure 3, it could be that toluene and ethylbenzene coeluted closely to the void time. Thus, these two compounds with very weak retentions are not suitable for the repeated analyses. For example, the similar time of ethylbenzene in all the columns may be just the results that they all have the similar void time. The authors should the compounds with the stronger retention such as pentyl benzene or better use the target analyte (CoQ10) in the reproducibility test. Otherwise, please provide the experimental evidence to address my concern here such as that with the raw chromatograms showing both retention times of ethylbenzene and thiourea void marker in all the cases.

At least, the %RSD results for the pentyl benzene or CoQ10 should be provided.

We apologize for this. Kindly see Fig 3 in the revised manuscript.